# Synthesis of Gentamicin-Immobilized Agar with Improved Antibacterial Activity

**DOI:** 10.3390/polym14152975

**Published:** 2022-07-22

**Authors:** Tingting Hou, Xin Wen, Lici Xie, Qixiang Gu, Chengpeng Li

**Affiliations:** School of Chemistry and Environmental Science, Guangdong Ocean University, Zhanjiang 524088, China; 18025076383@163.com (X.W.); xielici@126.com (L.X.); gdguqixiang@163.com (Q.G.); lichengpeng@gdou.edu.cn (C.L.)

**Keywords:** agar, gentamicin, antibacterial activity, gelling temperature, melting temperature

## Abstract

To develop agar derivatives with good antibacterial activity and decreased gelling and melting temperatures, two agar–gentamycin conjugates with 9.20% and 12.68% gentamicin immobilized were fabricated by oxidation, Schiff base and reduction reaction, and characterized by a Fourier Transform Infrared Spectrometer, ^1^H nuclear magnetic resonance and an elemental analyzer. It was found that the modifications changed the intermolecular interactions, leading to decreased gelling and melting temperatures for the oxidized agar and slightly increased gelling and melting temperatures for agar–gentamycin conjugates. Further studies of antimicrobial properties showed that the two agar–gentamycin conjugates possessed good antibacterial activity, which was positively correlated with the dosage and the immobilization rate of gentamicin. The minimum inhibitory concentration (MIC) and minimum bactericidal concentration (MBC) of agar–gentamycin conjugates with higher immobilization rates of gentamicin against *Escherichia coli* were 39.1 μg/mL and 156.2 μg/mL, respectively, and the MICs and MBCs against *Staphylococcus aureus* were 19.5 μg/mL and 78.1 μg/mL, respectively. A biofilm test indicated that certain concentrations of agar–gentamycin conjugate could effectively inhibit the biofilm formation of *Escherichia coli* and *Staphylococcus aureus*. In summary, agar–gentamycin conjugates possess good antibacterial activities and may be applied as a new kind of antibacterial material.

## 1. Introduction

Agar (AG) extracted from red algae (Rhodophyta) is a kind of gel-forming polysaccharide, which is extensively used in pharmaceutical, cosmetic and biotechnological industries [1]. AG is mainly composed of agarose and agaropectin. The former is a neutral polysaccharide with high gel strength, and the latter is sulphated polysaccharide with low gel strength [2]. Its gelling mechanism is due to the conformational transition (coil-to-helix) at low temperature and inter helical aggregation [3]. The substituents such as sulphate esters may cause kinks in the agar helix formation, hindering gel network formation and leading to low gel strength [4]. Normally, the gelling temperature and melting temperature of native agar is very high, which greatly limits its applications [5]. Therefore, a couple of chemical modifications such as methylation [6], oxyalkylation [5] and esterification [7], etc., have been used to decrease its gelling and melting temperature. For instance, modification using octenyl succinic anhydride can decrease the gelling temperature of AG by about 5 °C [7].

On the other hand, polymer-based wound dressings have been widely applied in various wounds. Due to good biocompatibility, biodegradability and hydrophilicity, AG has been used as wound dressings such as porous scaffolds [8], films [9], nonwoven fiber [10] and hydrogel [11]. However, AG possesses no antibacterial activity. To endow AG with antibacterial activity, antibiotics ciprofloxacin hydrochloride [12] and chloramphenicol [13] have been physically embedded in AG-based films and hydrogels, respectively, which can effectively inhibit the model pathogenic bacteria. Recently, amino groups have been selectively introduced on AG backbone, showing bactericidal and bacteriostatic activity for *P. aeruginosa*, *Escherichia coli* (*E. coli*) and *Staphylococcus aureus* (*S. aureus*) [14]. However, an antibiotics conjugation strategy has not been reported for AG functionalization. In our previous report, gentamicin-immobilized chitosan (CS-GT) showed greatly enhanced antibacterial activity for *Vibrio parahaemolyticus, Escherichia coli* and *Staphylococcus aureus* [15]. Herein, a new kind of gentamicin-immobilized agar (AG-GT) derivative was synthesized using our previous method in this article. The antibacterial performances of AG-GT were systematically analyzed, and the effects of the modification on gelling and melting temperatures were also evaluated. These results are expected to offer useful information for the development of antibacterial AG derivatives.

## 2. Materials and Methods

### 2.1. Materials

Agar, gentamicin sulfate (AR), ethylene glycol (AR), sodium cyanoborohydride and sodium periodate (AR) were all purchased from Shanghai Macklin Biochemical Co., Ltd. (Shanghai, China). *E. coli* (CMCC(B) 44103) and *S. aureus* (ATCC 25923) were all obtained from China Center of Industrial Culture Collection, Beijing, China. Tryptic soy agar (TSA) and Tryptone soya broth (TSB) were obtained from Beijing Land Bridge Technology Co., Ltd. (Beijing, China).

### 2.2. AG-GT Synthesis

CS-GT was prepared according to our previous method with slight modifications [16]. In detail, 2.00 g of AG was firstly dissolved in 400 mL of distilled water under a 95 °C water bath, which was then mixed with a 50 mL sodium periodate solution (18.8 wt%). The oxidation reaction was allowed to react at 45 °C in the dark for 6 h. Ethylene glycol (8.201 g) in 20 mL of water was then introduced to reduce the excessive sodium periodate in the dark for 6 h. The reaction solution was then dialyzed in a dialysis bag (molecular weight cut-off: 8 kD) at 50 °C for 2 days. After lyophilization (freeze dryer, Beijing Sihuan Scientific Instrument Factory, Beijing, China), the sample was collected and named as oxidized agar (OAG).

A 20 mL GT solution (10 wt%) was added to a 400 mL OAG solution (0.5 wt%) solution, and the reaction was carried out at 60 °C for 4 h. A 10 mL sodium cyanoborohydride solution (10 wt%) was then introduced and the reaction was kept at 50 °C for another 2 h. After dialysis at 50 °C for 2 days and lyophilization, the sample was collected and named as AG-GT-1. To explore the effects of GT dosage, AG-GT-2 was also synthesized using the same procedure with a 40 mL GT solution (10 wt%).

### 2.3. Characterization

The Fourier Transform Infrared Spectrometer (FTIR) spectrum was recorded using a Fourier Transform Infrared Spectrometer (TENSOR 27, Bruker, Karlsruhe, Germany) in the range of 4000 and 600 cm^−1^ using the attenuated total reflection mode with a resolution of 4 cm^−^^1^. The ^1^H nuclear magnetic resonance (^1^H NMR) spectrum was recorded on an AVANCE III 400 MHz superconductive Fourier Transform NMR spectrometer (Bruker, Switzerland) using D_2_O as a solvent.

The carbon, oxygen and nitrogen contents for all samples were determined on a Elementar Vario (EL cube, Heraeus, Germany) using Dumas’s combustion method. All forms of nitrogen and carbon were converted into N_2_ and CO_2_, respectively, which were then separated through chromatographic columns and detected using a thermos-conductive detector. All tests were repeated in triplicate, and an average was adopted. During the test, aspartic acid and urea were used as standards. The immobilized percentage (*I_p_*) of AG-GT-1 (or AG-GT-2) was calculated based on the following formula:(1)$$I_{p}=\frac{Nc}{12.80\%}\times 100\%$$
where *Nc* is the nitrogen content in sample AG-GT-1 and AG-GT-2, and 12.80% is the nitrogen content in GT.

### 2.4. Gelling and Melting Temperatures

The gelling and melting temperatures of AG and its derivatives were measured using the method reported by Chen and co-authors [2] with minor modification. For details, a 10 mL homogeneous AG-GT-1 solution (4 wt%) was transferred into a test tube and maintained isothermally at 45 °C in a thermostatic water bath for twenty minutes. The water bath was then cooled at a rate of 0.5 °C min^−1^, and the gelling temperatures were recorded by monitoring the temperature at which a nonrecoverable hole was formed when a glass rod was removed. Hot AG-GT-1 solutions (4 wt%, 10 mL) were immersed in a thermostatic water bath at 45 °C, which was then cooled to 10 °C at a rate of 0.5 °C min^−1^ for gelling. A glass bead (5 mm in diameter) was placed on the gel surface, and the temperature was raised from 10 °C to 100 °C at a rate of 1.5 °C/min. The melting temperature was recorded when the bead dropped to the bottom. For comparison, AG, OAG and AG-GT-2 were also tested using the same procedures. All experiments were performed in triplicate, and the average was used as the final result.

### 2.5. Antibacterial Assays

Two pathogenic bacteria *S. aureus* and *E. coli* were selected to evaluate the antibacterial activity for the AG-GT-1 and AG-GT-2 samples. For inhibition zone tests, 100 μL of bacterial suspension at 10^6^~10^7^ colony-forming units (CFU) mL^−^^1^ was spread on the top of nutrient agar plates. A 10 μL (5 mg/mL or 10 mg/mL) sample solution was dipped on the filter sheets with a diameter of 6 mm, which were then placed on the agar plate. After incubation for 24 h at 37 °C, the growth inhibition zones were measured using a vernier caliper. For comparison, the inhibition zones of GT were also tested using the same procedure. The final results were averaged based on three parallel experiments.

The minimum inhibitory concentration (MIC) was determined via a microdilution method. Ten mL of standardized suspension in the range of 10^6^ and 10^7^ CFU mL^−1^ was transferred to the above tubes. The tube containing only bacterial suspension was defined as a positive control group, while the tube with only TSB medium was defined as a blank control group. After incubation in the presence of control or different concentrations of samples (i.e., GT, AG-GT-1 and AG-GT-2) at 37 °C for 18 h, the lowest sample concentration with no visible growth of bacteria was determined as the MIC. The minimum bactericidal concentration (MBC) was determined using the spread plate method. A 50 μL bacterial suspension without visible growth above was transferred to the agar plates and incubated for 24 h at 37 °C. The lowest concentration which could kill 99.9% of the starting inoculum was defined as the MBC. Both MIC and MBC tests were repeated in triplicate, and an average was recorded as the final results.

The effect of AG-GT-1 or AG-GT-2 on preventing biofilm formation was evaluated as follows: 2% TSA was transferred to a flat-bottom 24-well polystyrene microtiter plate (Corning, Cambridge, MA, USA). After gelling, a bacteria-free cover glass (10 mm × 10 mm) was inserted in each well vertically. A 100 μL bacterial suspension at 10^6^~10^7^ CFU mL^−1^ and 1.9 mL TSB with a sample (i.e., AG-GT-1 and AG-GT-2) concentration in the range of 1/4 MIC and 4 MIC was added to a set of wells. A 100μL bacterial suspension (10^6^~10^7^ CFU mL^−1^) and 1.9 mL of TSB were also added and used as a blank control. All plates were incubated at 37 °C for one, three or five consecutive days. To produce robust biofilms, 1 mL of fresh TSB with a predetermined concentration of AG-GT-1 or AG-GT-2 was changed every day.

After a predetermined incubation time, biofilms were taken out and washed thoroughly using PBS solution to remove any loosely attached bacterial cells. Bacteria within the biofilm attached to the wells were then stained using methylene blue solution (0.01 wt%). After treatment for fifteen minutes, surface methylene blue was removed by rinsing with sterile sodium chloride solution (0.9%). The representative samples were then selected for imaging using an optical microscope (MP41, Mingmei, Guangzhou, China).

## 3. Results and Discussion

### 3.1. Characterization

#### 3.1.1. Chemical Structure

As can be seen from Figure 1a, the broad peak centered at 3356 cm^−1^ was attributed to the stretching vibration of the hydroxy group in AG. The multiple peaks in the range of 3000 and 2800 cm^−1^ were the stretching vibration of CH_2_ in AG. The peak centered at 1620 cm^−1^ was the signal of amide I vibrations, indicating the presence of proteins [4]. After oxidation, a small shoulder peak at 1675 cm^−1^ appeared, showing the aldehyde groups were produced during oxidation. This shoulder peak became invisible after the immobilization -of GT, which was possibly due to the formation of a C-N group. ^1^H NMR spectra were then recorded to further analyze the structure changes during the reaction in Figure 1b, and the multiple signals between 4.0 and 5.2 ppm were attributed to the signals in AG [9]. After oxidation, the weak peaks located at 8.37 ppm and 9.3 ppm were due to the hydrogen signals from the aldehyde group and carboxyl group generated, respectively [17]. As expected, lots of the characteristic signals in GT also emerged in AG-GT-1 and AG-GT-2, indicating GT was successfully immobilized. According to Table 1, the nitrogen contents in AG and OAG were 0.16% and 0.20%, indicating trace protein existed in AG and OAG, which coincided well with the results in FTIR. Compared to AG, the oxygen content in OAG was improved and carbon content in OAG was decreased, which may have been due to the oxidation. The improved nitrogen content in OAG indicated that those nitrogen-containing functional groups in protein remained after the oxidation, while carbon-containing functional groups may have been partly decomposed into small molecules. On the other hand, the nitrogen contents of AG-GT-1 and AG-GT-2 improved significantly after the covalent linkage of the GT. Based on Formula (1), *I_p_* for AG-GT-1 and AG-GT-2 were 9.20% and 12.68%, respectively, demonstrating that a higher GT dosage leads to a higher immobilized percentage.

#### 3.1.2. Gelling and Melting Temperatures

As can be seen from Figure 2a, the gelling temperatures of the modified samples (i.e., OAG, AG-GT-1 and AG-GT-2) all decreased, and the lowest gelling temperature was only 32.90 ± 0.13 °C for OAG. It was interesting to find that the gelling temperature of AG-GT-1 and AG-GT-2 showed a slight increased inclination after further immobilization of GT. Meanwhile, the melting temperatures showed the same inclination as that of the gelling temperatures. According to the references, this may be due to the decreased intermolecular interactions due to the oxidation [18]. It was worth mentioning that further GT immobilization and reduction may lead to the slightly enhanced intermolecular interactions.

### 3.2. Inhibition Zone

As can be seen from Figure 3, Figure 4 and Table 2, a higher dosage was helpful for antibacterial performance, and all samples were more active in inhibiting *E. coli* than *S. aureus*. As expected, the immobilization of GT led to the greatly improved antibacterial activities. Due to a higher immobilization ratio of AG-GT-2, the antibacterial activity of AG-GT-2 was better than that of AG-GT-1.

### 3.3. MIC and MBC

Table 3 shows that the MIC and MBC of GT for *Escherichia coli* were 7.81 μg mL^−1^ and 15.63 μg mL^−1^, and the MIC and MBC for *S. aureus* were 3.91 μg mL^−1^ and 7.81 μg mL^−1^, respectively, which were all significantly lower than those of AG-GT-1 and AG-GT-2. Compared to AG-GT-1, the MIC and MBC of AG-GT-2 for both *Escherichia coli* and *S. aureus* were much lower. These results again demonstrated that a higher immobilization ratio of gentamicin was beneficial for antibacterial activity.

### 3.4. Biofilm Inhibition

As can be seen from Figure 5 and Figure 6, *E. coli* biofilms were built after incubation for 1, 3 or 5 days, and their cell densities were positively related to the incubation time in the control group. After the treatment of AG-GT-1 and AG-GT-2, the cell density decreased significantly. In the low-concentration groups (i.e., 0.25 MIC, 0.5 MIC and MIC), thick biofilms with dense cells were still notably visible. In high-concentration groups (i.e., 2 MIC and 4 MIC), only very thin biofilms with large empty spaces were found, showing strong inhibition activity. This inhibitory effect showed an enhanced inclination with the prolonged treatment time in high-concentration groups. Similarly, low-concentration groups (i.e., 0.25 MIC, 0.5 MIC and MIC) could only slightly inhibit the formation of *S. aureus* biofilms, and high-concentration groups (i.e., 2 MIC and 4 MIC) could effectively inhibit the formation of *S. aureus* biofilms (Figure 7 and Figure 8). Both *E. coli* and *S. aureus* biofilms, however, could not be fully inhibited even at a treated concentration of 4 MIC for AG-GT-1 or AG-GT-2.

## 4. Conclusions

FTIR and ^1^H NMR spectra analysis showed that two agar–gentamycin conjugates, AG-GT-1 and AG-GT-2, were successfully synthesized via oxidation, Schiff base and reduction reaction. Elemental analysis indicated that the immobilization percentages for AG-GT-1 and AG-GT-2 were 9.20% and 12.68%, respectively. Further analysis found that the oxidation reaction resulted in significantly decreased melting and gelling temperatures. Schiff base and reduction reactions, however, led to slightly increased melting and gelling temperatures. The inhibition zone test found that AG-GT-1 and AG-GT-2 possessed good antimicrobial activities, which were positively correlated with the dosage of antibacterial agents and immobilization rate of gentamicin. Under the experimental conditions, the MIC and MBC of AG-GT-2 were only about 25% of those of AG-GT-1. Biofilm tests found that low-concentration groups (MIC or less) could only slightly inhibit the biofilm formation for *S. aureus* and *E. coli*, whilst high-concentration groups (2 MIC and 4 MIC) could effectively inhibit the biofilm formation for *S. aureus* and *E. coli*. To summarize, agar–gentamycin conjugates with decreased melting and gelling temperatures showed superior antimicrobial activities. Further study will focus on their potential applications in infected wounds.

## Figures and Tables

**Figure 1 polymers-14-02975-f001:**
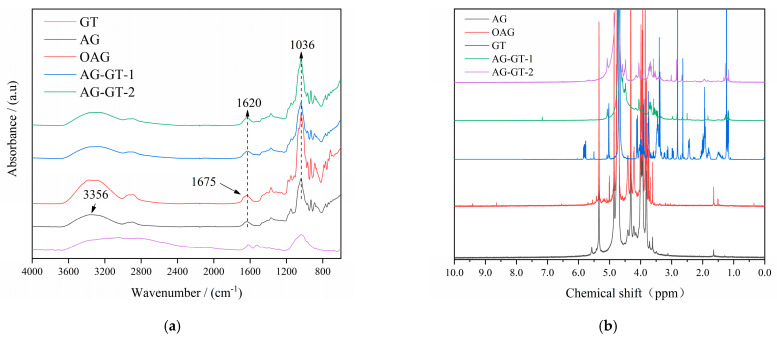
FTIR spectra (**a**) and ^1^H NMR spectra (**b**) for GT, AG, OAG, AG-GT-1 and AG-GT-2.

**Figure 2 polymers-14-02975-f002:**
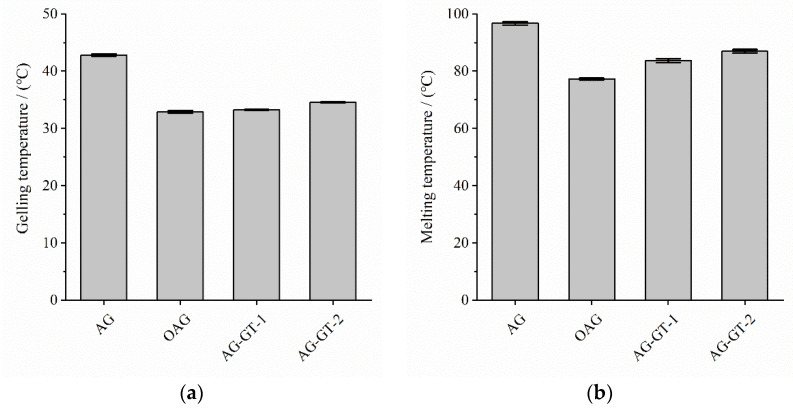
Gelling temperature (**a**) and melting temperature (**b**) for AG and its derivatives.

**Figure 3 polymers-14-02975-f003:**
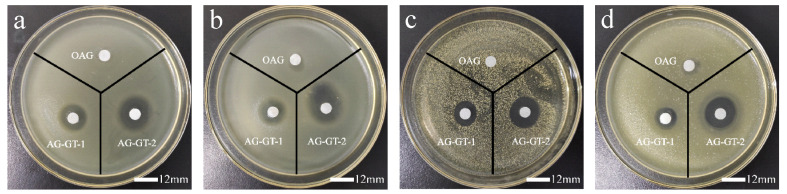
Inhibition zone of *E. coli* treated with sample solutions at a concentration of 5 mg/mL (**a**) and 10 mg/mL (**b**); inhibition zone of *S. aureus* treated with sample solutions at a concentration of 5 mg/mL (**c**) and 10 mg/mL (**d**).

**Figure 4 polymers-14-02975-f004:**
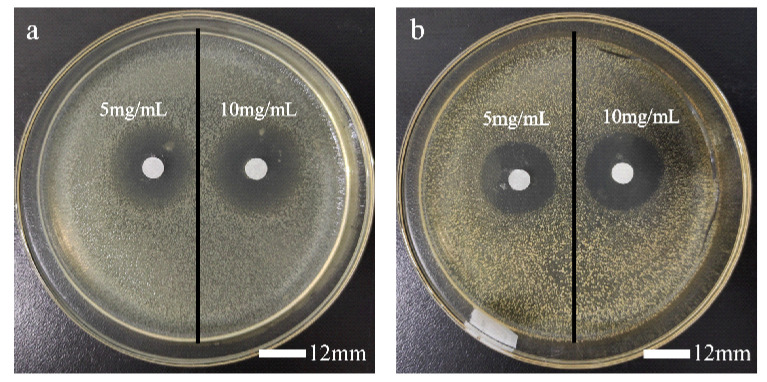
Inhibition zone of *E. coli* (**a**) and *S. aureus* (**b**) treated with GT.

**Figure 5 polymers-14-02975-f005:**
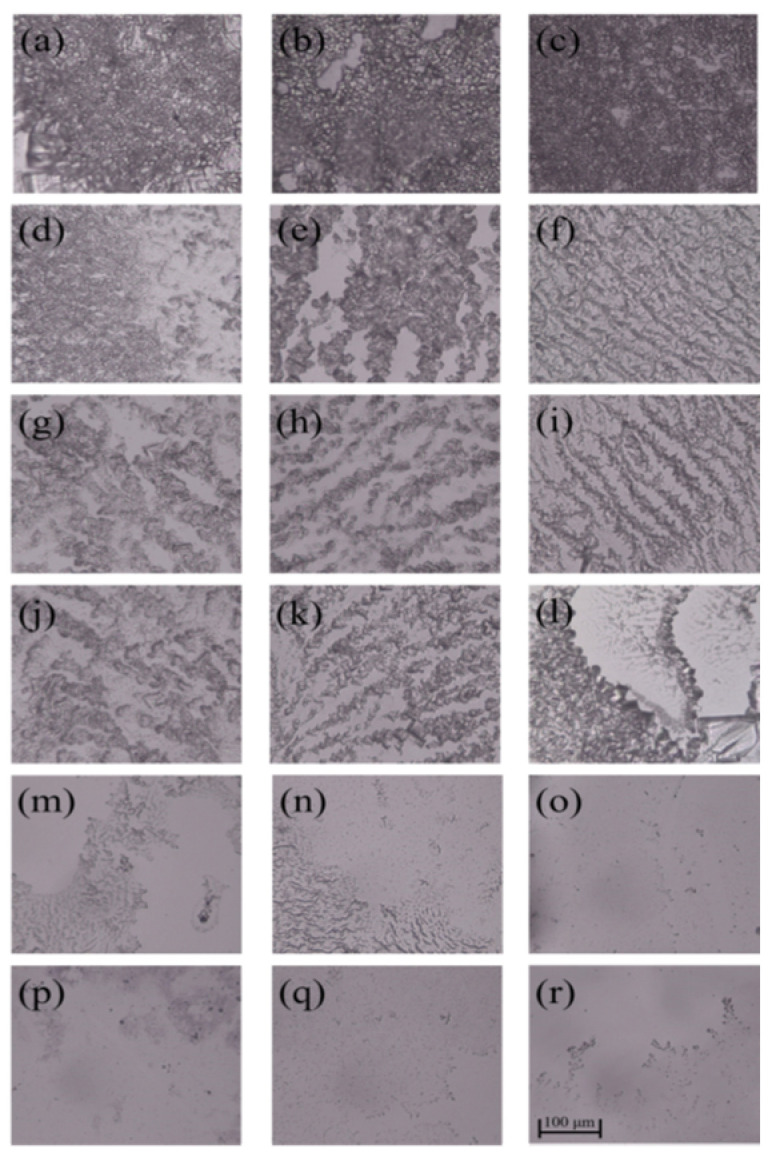
The inhibitory effect of AG-GT-1 on *E. coli* biofilms: (**a**–**c**) are the *E. coli* biofilms developed for 1, 3 and 5 days in control group, (**d**–**f**) are the *E. coli* biofilms treated with AG-GT-1 at a concentration of 1/4 MIC for 1, 3 and 5 days, (**g**–**i**) are the *E. coli* biofilms treated with AG-GT-1 at a concentration of 1/2 MIC for 1, 3 and 5 days, (**j**–**l**) are the *E. coli* biofilms treated with AG-GT-1 at a concentration of MIC for 1, 3 and 5 days, (**m**–**o**) show the *E. coli* biofilms treated with AG-GT-1 at a concentration of 2 MIC for 1, 3 and 5 days, (**p**–**r**) present the *E. coli* biofilms treated with AG-GT-1 at a concentration of 4 MIC for 1, 3 and 5 days, respectively.

**Figure 6 polymers-14-02975-f006:**
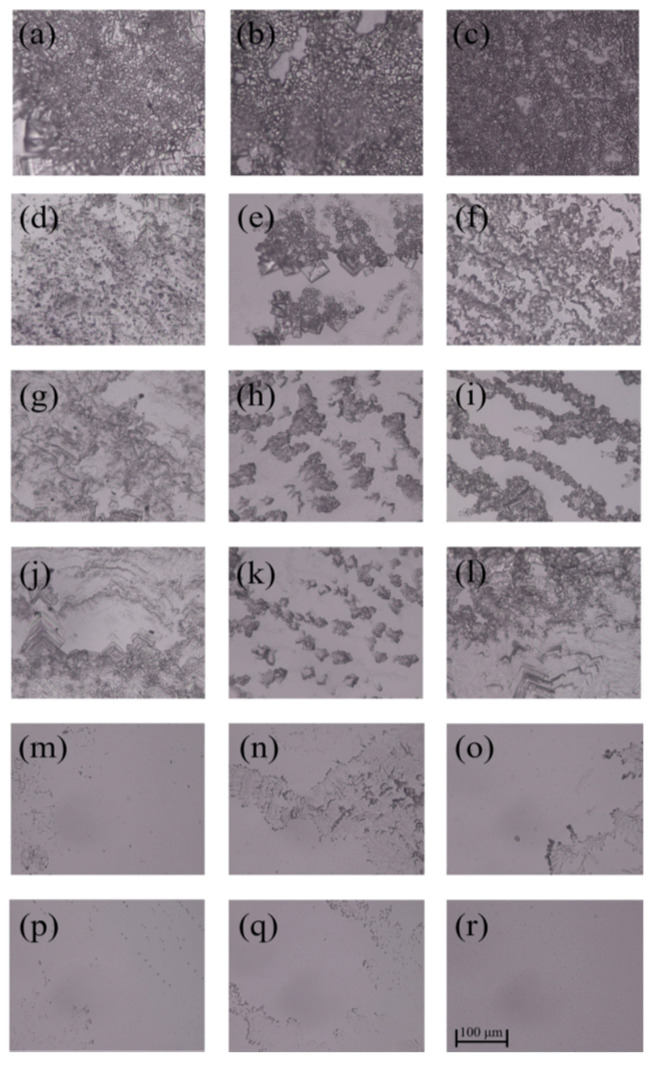
The inhibitory effect of AG-GT-2 on *E. coli* biofilms: (**a**–**c**) are the *E. coli* biofilms developed for 1, 3 and 5 days in control group, (**d**–**f**) are the *E. coli* biofilms treated with AG-GT-2 at a concentration of 1/4 MIC for 1, 3 and 5 days, (**g**–**i**) are the *E. coli* biofilms treated with AG-GT-2 at a concentration of 1/2 MIC for 1, 3 and 5 days, (**j**–**l**) are the *E. coli* biofilms treated with AG-GT-2 at a concentration of MIC for 1, 3 and 5 days, (**m**–**o**) show the *E. coli* biofilms treated with AG-GT-2 at a concentration of 2 MIC for 1, 3 and 5 days, (**p**–**r**) present the *E. coli* biofilms treated with AG-GT-2 at a concentration of 4 MIC for 1, 3 and 5 days, respectively.

**Figure 7 polymers-14-02975-f007:**
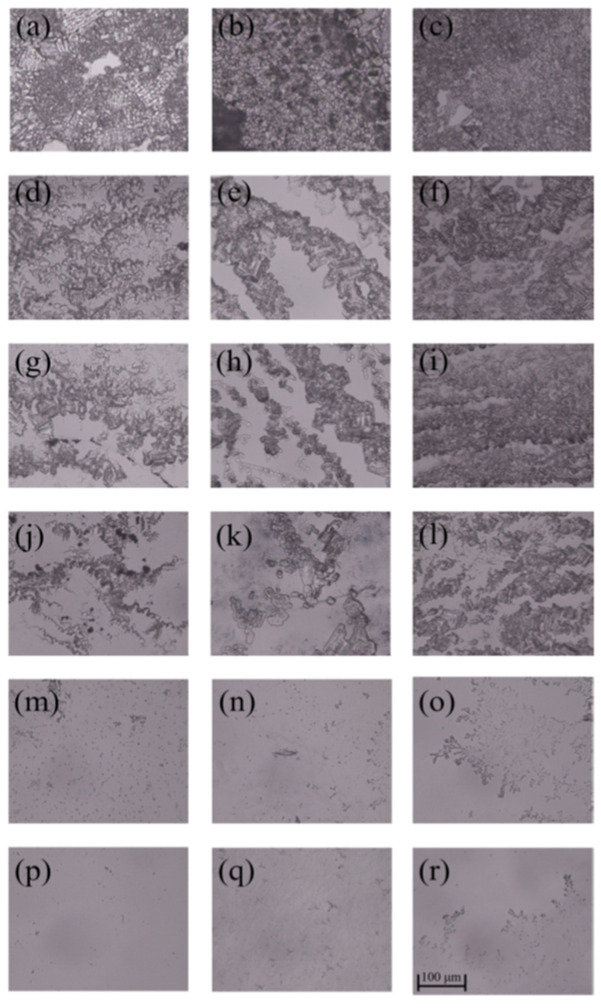
The inhibitory effect of AG-GT-1 on *S. aureus* biofilms: (**a**–**c**) are the *S. aureus* biofilms developed for 1, 3 and 5 days in control group, (**d**–**f**) are the *S. aureus* biofilms treated with AG-GT-1 at a concentration of 1/4 MIC for 1, 3 and 5 days, (**g**–**i**) are the *S. aureus* biofilms treated with AG-GT-1 at a concentration of 1/2 MIC for 1, 3 and 5 days, (**j**–**l**) are the *S. aureus* biofilms treated with AG-GT-1 at a concentration of MIC for 1, 3 and 5 days, (**m**–**o**) show the *S. aureus* biofilms treated with AG-GT-1 at a concentration of 2 MIC for 1, 3 and 5 days, (**p**–**r**) present the *S. aureus* biofilms treated with AG-GT-1 at a concentration of 4 MIC for 1, 3 and 5 days, respectively.

**Figure 8 polymers-14-02975-f008:**
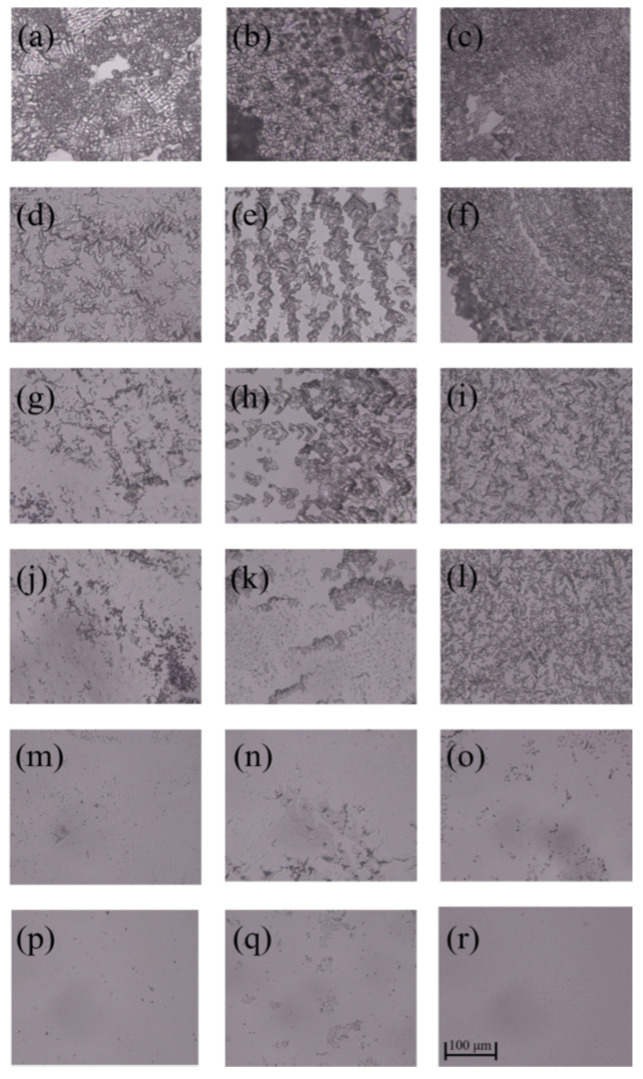
The inhibitory effect of AG-GT-2 on *S. aureus* biofilms: (**a**–**c**) are the *S. aureus* biofilms developed for 1, 3 and 5 days in control group, (**d**–**f**) are the *S. aureus* biofilms treated with AG-GT-2 at a concentration of 1/4 MIC for 1, 3 and 5 days, (**g**–**i**) are the *S. aureus* biofilms treated with AG-GT-2 at a concentration of 1/2 MIC for 1, 3 and 5 days, (**j**–**l**) are the *S. aureus* biofilms treated with AG-GT-2 at a concentration of MIC for 1, 3 and 5 days, (**m**–**o**) show the *S. aureus* biofilms treated with AG-GT-2 at a concentration of 2 MIC for 1, 3 and 5 days, (**p**–**r**) present the *S. aureus* biofilms treated with AG-GT-2 at a concentration of 4 MIC for 1, 3 and 5 days, respectively.

**Table 1 polymers-14-02975-t001:** Elementary mass percentage for AG, OAG, AG-GT-1 and AG-GT-2.

Samples	C (%)	N (%)	O (%)
AG	38.24	0.16	56.38
OAG	29.65	0.20	64.24
AG-GT-1	37.33	1.35	55.23
AG-GT-2	36.84	1.86	54.17

**Table 2 polymers-14-02975-t002:** Diameter of inhibition zone for GT, OAG, AG-GT-1 and AG-GT-2.

Samples	Concentrations (mg mL^−1^)	*E. coli* (mm)	*S. aureus* (mm)
GT	5	23.75 ± 0.93	23.01 ± 0.97
	10	25.75 ± 0.66	24.27 ± 0.55
OAG	5	7.12 ± 0.49	-
	10	8.58 ± 0.53	8.3 ± 1.37
AG-GT-1	5	14.32 ± 0.70	14.23 ± 0.28
	10	15.83 ± 0.94	15.55 ± 0.47
AG-GT-2	5	18.98 ± 0.80	17.98 ± 0.36
	10	20.26 ± 0.34	19.32 ± 0.87

**Table 3 polymers-14-02975-t003:** **Table 2** MIC and MBC of GT, AG-GT-1 and AG-GT-2.

Samples	*E. coli*		*S. aureus*	
	MIC (μg/mL)	MBC (μg/mL)	MIC (μg/mL)	MBC (μg/mL)
GT	7.81	15.63	3.91	7.81
AG-GT-1	156.2	625	78.1	312.5
AG-GT-2	39.1	156.2	19.5	78.1

## Data Availability

The data presented in this study are contained within the article.

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
