# Peer review of "Synthesis of Gentamicin-Immobilized Agar with Improved Antibacterial Activity"

_polymers, 2022, doi:10.3390/polym14152975_

Round 1

Reviewer 1 Report

The authors have prepared agar-gentamycin conjugates by oxidation, Schiff base and reduction reaction to improve their antibacterial activity. The authors have demonstrated agar-gentamycin conjugates with good antibacterial activity dur to positively correlated with the grafting rate of gentamicin. Overall, this work can inspire more material design ideas of agar-gentamycin conjugates for antibacterial application. Therefore, I would like to recommend this work to publish in Polymers. Below are some comments for the authors.

1.      The abbreviations should be explained for the first-time use. For example, in the abstract, the abbreviations of MIC and MBC should be explained.

2.      The section of “Conclusions” is too short. More important information about this study should be described in “Conclusions”.

3.      The section “Introduction” is too less. The authors should provide some information related with this work in the abstract. What did that do for this work?

Author Response

Thank you for your constructive suggestions. Please see attched Response and revised manuscript for details.

Reviewer 2 Report

Dear Authors

The results presented in your work are very interesting for the readers, regardless of the unclear novelty in which the same idea has been investigated by you in previously published work.

1-I have the main concern about the use of "Grafting" terminology describing the covalent linkage of the Gentamicin molecules to the agar backbone.

I believe that using of "immobilized" terminology is more accurate especially when the authors described the modified agar withGentamicin by conjugation process.

2- The authors did not performe any optimization process of the oxidation conditions. Moreover, No clear data about the oxidation degree or even the exact ammount of immobilized Gentamicin were given.

The authors need to take the above mentioned comments into consideration before thier work can be considered for publication.  

Author Response

Thank you for your helpful suggestions. We have made corresponding corrections, please see attached response and revised manuscript for details.

Round 2

Reviewer 1 Report

The authors have addressed all issues raised by the reviewers.

Reviewer 2 Report

Dear Authors

The revised version of your manuscript has been improved according to the comments taken into consideration.

Accordingly, I can accept the current form of your revised manuscript for publication.